# Impact of feeding strategies on the welfare and behaviour of horses in groups: An experimental study

Marie Roig-Pons[1,2,3]*, Iris Bachmann[2], Sabrina Briefer Freymond[2]

**1** Animal Welfare Division, Veterinary Public Health Institute, University of Bern, Bern, Switzerland, **2** Agroscope, Swiss National Stud Farm, Avenches, Switzerland, **3** Graduate School for Health Sciences, University of Bern, Bern, Switzerland

\* marie.roigpons@gmail.com

## Abstract

Finding feeding strategies that meet horses' needs without compromising health is essential for optimising welfare, particularly in group housing, where limited hay availability increases aggression and injury risks. Recently, two strategies have emerged: portioning daily intake into smaller, frequent meals using time-controlled hay racks, or slowing intake with "slow-feeders." However, the effects of such management practices on horse behaviour remain underexplored. We conducted a cross-over study with 18 mares divided into four groups to compare three feeding strategies: "traditional" (3 of 2-hours meals during daylight, TD), "portioned" (6 of 1-hour meals spread over 24h, PO) and "slow-feeding" (ad libitum hay covered by a net, SF). Each treatment included 3 weeks of habituation and 2 weeks of data collection. We continuously recorded social interactions for 15 hours and noted the position and activity of all horses every 15 minutes. We also recorded injuries periodically and measured the lying behaviour using accelerometers. We analysed the effects of treatment on agonistic and affiliative behaviour within groups using generalised mixed model and selected the best model using AIC. We used the same procedure for the injuries and lying behaviour at the individual level. Horses in SF exhibited activity time budgets resembling natural conditions, while TD and PO resulted in time budgets similar to box-stall systems, despite the loose-housing system. Surprisingly, our results suggest that PO may be more frustrating for the horses than TD. Indeed, there was no significant reduction of agonistic behaviours during feeding times in PO compared to TD and lying behaviour tended to be impaired in PO (−11.3 min/day, 95% CI [−25.8; 3.1]) compared to SF and TD (37.5 min/day on average). In our study, portioning into smaller, more frequent meals did not reduce the stress in horses. This highlights the need for further research on portioning strategies to find optimal feeding management. In addition, slow-feeding was a more suitable feeding strategy for horses than

**Data availability statement:** All relevant data for this study are publicly available from the Zenodo repository (https://doi.org/10.5281/zenodo.15090821).

**Funding:** This research was conducted as part of a PhD thesis: Roig-Pons, M. (Horses and slow-feeders: investigating consequences on horse health and behaviour), Animal Welfare Division, University of Bern, Bern, Switzerland, in collaboration with Agroscope.

**Competing interests:** The authors have declared that no competing interests exist.

portioning. However, more research is required to substantiate the initial findings on the efficacy of ad libitum slow-feeding on the horse's health and behaviour.

## Introduction

Horses are highly social animals. Under natural conditions, feral horses live in family groups called "herds" comprising several mares with their offspring, usually with one or two stallions or in small groups of stallions called "bachelors" [1–3]. Individuals can remain in these groups for several years [2,4,5]. Social interactions represent only a small proportion of a horse's activity time-budget (3–4%), yet they are essential for the stability and cohesion of the equine social unit [6,7]. The presence of conspecifics provides not only opportunities for social interaction but also a sense of security, synchronization of rhythms, social learning for young horses, and protection from predators [3,4,7–9]. Bonds between feral horse mares have been shown to enhance fitness and reproductive success [10]. However, domestication and captivity have led to social isolation for many horses, which negatively impacts their psychological health [11]. Isolated horses are more likely to develop abnormal, repetitive behaviours [12–15]. Despite their social nature and the adverse effects of isolation, many horses are still housed individually due to concerns about injuries in group housing. Identifying solutions to reduce injury risk in group housing is therefore essential to promoting better welfare for horses.

Group housing is recommended by many authorities, yet individual housing remains common, especially in some countries [11,16]. Practicality, space limitations, building organization, ease of handling, and tradition are key reasons for this preference. Consequently, individual housing is still prevalent [11]. Owners also often cite the risk of injury, particularly for high-value sport horses, as a major concern [17]. In Switzerland, Dittmann *et al.* [18] found that 15% of competition horses were housed in groups, compared to 43% of leisure horses. However, the level of aggression within groups is low in natural conditions [3,19]. Therefore, the risk of injury appears to be associated with the conditions in which the animals are kept and management practices [11,20]. The availability of resources, the size and stability of the group, and the social experiences of the horses can all contribute to the emergence of aggressive behaviour [20–23]. However, this also means that for a given group in a given space, optimal feeding management can help prevent aggression events.

In addition to their social needs, horses are herbivores with a constant secretion of acid in their stomachs and need to eat regular and small meals to avoid digestive problems [24–26]. Under natural conditions, they spend over 70% of their time foraging or feeding and can become bored or frustrated if they are unable perform this behaviour [27]. As a result, increasing the duration of daily availability of hay is beneficial for the horses' health and it regulates the level of aggressiveness between horses, thus reducing the risk of injuries [23,28,29]. The provision of *ad libitum* hay to group-housed horses has been shown to halve the incidence of agonistic interactions in comparison to situations where horses have no access to hay [28]. However, this

is not always feasible due to practical, financial, and health concerns. Many horses are "easy keepers" with low energy expenditure or metabolic predispositions, making them prone to obesity when fed *ad libitum* [30–33]. Obesity can lead to serious issues such as musculoskeletal overload, immune system alterations, and diseases like laminitis [33,34]. Therefore, it is crucial to find feeding practices that prevent aggression without compromising horse health.

Two primary feeding strategies have been developed to meet horses' needs without compromising their body condition: portioning intake into smaller, frequent meals and slowing down intake. The first strategy uses time-controlled hay racks, which open and close at set times. This allows frequent feeding and reducing fasting periods without increasing the amount of feed ingested and the caretaker's workload. The second strategy involves "slow-feeders," designed to extend hay intake time [35,36]. These come in various forms, such as plastic boxes with grids, suspended bags with small holes, and racks with closely spaced bars, with hay nets being the most common [37]. By making hay less accessible, horses spend more time eating the same amount of hay. Studies have shown that hay nets increase feed intake time [35,38], but little is known about the effects of such management practices on the behaviour of horses kept in groups.

While slow-feeding and portioning strategies appear promising, few studies have assessed their effects on the welfare of group-housed horses [29,39]. It is unclear how these strategies impact group dynamics, reduce frustration, and lower injury risks compared to traditional feeding. Recent research suggests slow-feeding may enhance welfare more than portioned feeding, with similar hay intake results [29]. However, no experimental studies have compared these strategies to traditional feeding methods (2–3 forage meals per day).

This study aimed to evaluate two feeding strategies for optimising horse nutrition while minimizing weight gain. Additionally, we compared two restrictive feeding regimes—portioned and traditional feeding—to determine if portioning improves horse welfare. We hypothesised that slow-feeding (*ad libitum* with a net) would enhance welfare by promoting a natural time budget, reducing aggression, and increasing affiliative interactions, though possibly causing frustration with the net. We also hypothesised that portioned feeding would improve welfare over traditional feeding by reducing agonistic interactions during feeding times due to more frequent feeding bouts. However, we expected the time budgets for the two restrictive treatments to be similar, given the same total hay accessibility.

## Materials and methods

The study was conducted between February and July 2023, at the experimental site of Agroscope, Swiss national stud farm (SNSF).

### Animals and housing

The experiment included 18 mares, randomly divided into 4 groups (two groups of five and two groups of four individuals). The allocation was stratified to ensure the best possible barefoot/shoe ratio (2/2 or 2/3) for a second study conducted in parallel. Mares were accustomed to their group for 7.5 months prior to the study. They were moved from their original loose group-housing system at the SNSF to the new experimental site in January, 1.5 months before the start of the study. Mare ages ranged from 6 to 15 years old (median = 13 ± 2.4) and they were all warmbloods (Swiss warmblood or German warmblood). The mares were research mares only: they were not ridden and had little contact with humans, except for the experimenters and grooms who cleaned their area once a day. In terms of body condition, at the start of the experiment all the mares had a BCS between 4 and 7 (i.e., they were all in perfect to overconditioned according the Henneke scale [40]).

The housing at the experimental site consisted of four identical paddock-trails with two areas of interest: one for lying down (15 x 30m = 450 m²) and another one for feeding (20 x 30m = 600 m²), connected by two corridors, each 130m long and 5m wide (650 m²). However, for practical reasons, only one trail was opened during the course of the study. The feeding and lying down areas were stabilized with grids and a substructure, while the connecting trails were a mix of bare ground and stabilized soil covered with wood pellets. The lying down area consisted of a shelter 15m long and 5.5m wide (82.5m²), with a water trough on the opposite side. The feeding area consisted of a time-controlled hay rack with 16

feeding places (3.5 x 2.5m), with a rolling door that could be programmed to open and close at certain times. After the first repetition, a rolling area (squared area of 4.70m x 4.70m, filled with sand) was also added to the feeding area to encourage lying down behaviour. An aerial photograph of the experimental site, taken by a drone, is shown in Fig 1A.

### Feeding treatments

The horses were fed using a hay rack with a time-controlled feeding system. Only one half of the rack (eight feeding places) was open at any given time, with the open sides alternating (Fig 1B). The horses did not receive any feed concentrate, but they had access to a licking stone (one per group). The hay provided to the mares was first and second cut hay, sourced from meadows at an altitude of 500–700 metres. This hay had a relatively low energy content and was fibrous, with stalks measuring 15 centimetres±5 centimetres in length. The study employed a Latin square design with three repetitions (with data collection) and a repetition "0" (no data collection associated), to minimize carry-over effects. The detailed experimental plan can be found in S1 Table. The treatments included:

- Traditional (TD): Hay available in three 2-hour slots daily (7–9 am, 1–3 pm, 7–9 pm), totalling 6 hours.

- Portioned (PO): Hay available in six 1-hour slots daily (3–4 am, 7–8 am, 11–12 pm, 3–4 pm, 7–8 pm, 11–12 pm), also totalling 6 hours.

- *Ad libitum* with hay nets (Slow-feeding, SF): Hay accessible at all times with one side of the rack open and covered by a 40 mm mesh hay net.

### Data collection

For each repetition, data was collected over a period of two weeks after a three-week habituation period. The duration of the habituation period was determined based on the findings of Rochais *et al.* [36], who reported that the impact of the feeding treatment on the horse behaviour and time-budget differed between the initial two weeks and the third week. This suggests that at least two weeks are necessary for the horses to adapt to a new feeding regime. Given that the horses in Rochais's study were individually housed and fed, it was deemed necessary to implement an additional week to ensure

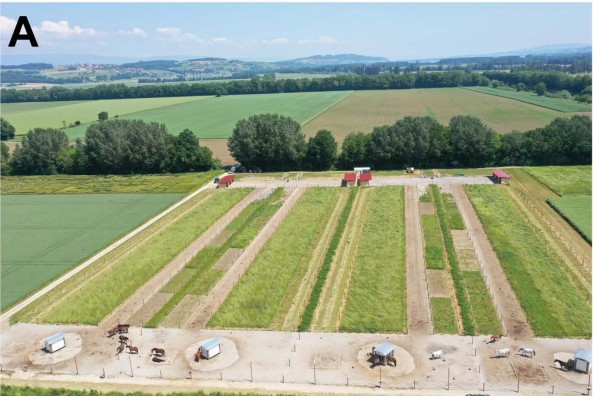 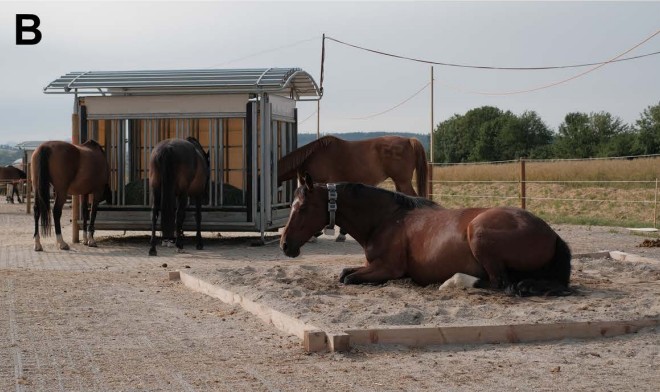

**Fig 1. Illustration of the experimental conditions in a feeding management study carried out on 18 mares, divided into four groups. A.** The experimental site: In the foreground are the time-controlled hay racks, with the sand areas to the left. A trail connects the feeding area with the resting area in the background (red building: shelters). Photograph courtesy of Marianne Cockburn. **B.** In the foreground the rolling area and in the background the hay racks used in the trials, with a total of 16 feeding places and a rolling door on each side (closed on the opposite sides at the time of the photograph).

a sufficient habituation time. The following description of the data collection is for one repetition and the same procedure was used for the three repetitions.

Note: Regarding body condition, we originally decided to rely on body weight changes, using a weighting scale. However, due to technical constraints relating to the margin of error of the weighing scale, the short duration of each feeding regime, as well as an unforeseen event (see section "Disruption from experiment"), body weight changes could not be accurately assessed over the study and was therefore removed from the analysis.

**Social interactions.** Our goal was to observe each group for a total of 15 hours, divided into 15 one-hour slots. For groups under the TD or PO treatments, we aimed to observe them for five hours during feeding times and 10 hours during non-feeding times. Additional observations during non-feeding times were conducted due to increased aggressiveness associated with limited resources [21,23,28], facilitating quicker detection of interactions between pairs.

Two experimenters, E1 and E2, conducted the observations. E1 had prior experience observing horses and cows from previous research. E2 underwent two weeks of training before starting. E1 trained E2 using video material of labelled social interactions, ensuring consistency in observational descriptions before both began field observations together. Observers were positioned outside of the group in order to avoid disruptions. Affiliative (positive social behaviours that strengthen social bonds and cohesion) and agonistic (behaviours related to conflict or competition, including aggressive and submissive actions) interactions were documented using a digital recorder with pauses between recordings. Observers marked fifteen-minute intervals to denote each quarter-hour segment. The ethogram used for affiliative interactions was adapted from Jørgensen *et al.* [41] and Heitor *et al.* [42], while the ethogram from Burla *et al.* [23] guided the classification of agonistic behaviours. A summary of the ethogram is available in Table 1 and the ethogram is further detailed in S2 Table. Routine disruptions like passing tractors or trail cleaning did not interrupt observations. However, major disruptions, such as a horse being removed from the group or caretakers feeding carrots nearby, halted observations until normal conditions resumed. Each observation session lasted one hour, ensuring each horse was observed at least once per time slot, except during night-time periods when observations were impractical. To maintain observation quality, observers took mandatory breaks after two consecutive hours of monitoring.

**Time-budget and spatial positioning.** Throughout continuous observations, scans were conducted every fifteen minutes (five scans per hour) to monitor various activities of the horses (feeding, foraging, standing, lying, resting, walking, interacting, and other behaviours). During the scans, the observers also recorded the specific zones where the horses were located (feeding areas, trails or shelter/trough areas).

**Injuries.** At the start of each data collection period, baseline injuries were documented, and existing injuries on the horses were recorded. Follow-up injury assessments were conducted on days +2 and +4, where any new injuries were added to the record scheme. This was done by one experimenter restraining a mare while a second experimenter

**Table 1. Summarised ethogram used during the observations of four groups of horses in an experimental study assessing the effect of feeding management on the welfare of horses housed in groups. Each group was observed for 15 hours per treatment, over three different treatments. The valence and type of interactions are also presented.**

| Valence of the interaction | Type of interactions | Behaviours included |
|---|---|---|
| Affiliative interactions | Movement | Follow |
| | Proximity | Approaches (followed by contact or standing rest nearby) [42] |
| | Approach for social interactions | Ask for grooming, ask for play |
| | Actions | Contact, Allo-grooming or Play |
| Agonistic interactions | Passive displacements | Passive displacement |
| | Push | Push |
| | Threatening behaviours | Back, Head threats, Threat to bite, Threat to kick |
| | Aggressive behaviours | Bite, Kick, Attack, Chase, |

assessed the horse. The mares were assessed in their group to avoid any stress to the horses. Each new injury was documented with its dimensions (length times width) and location in a table. Additionally, injuries were assigned a severity score based on a modified scale (continuous scale ranging from 1 for removed hair without skin lesions to 5 for severe injuries requiring surgery), derived from Zollinger *et al.* [43].

**Lying behaviour.** The lying behaviour of all horses was recorded using MSR145 data loggers (MSR Electronics GmbH, Seuzach, Switzerland). The devices were attached to the metacarpal bone of one of the front legs, as described by Burla *et al.* [23]. To prevent any injury to the leg, we used a home-made foam protection to attach the logger (see Fig 2). Only the data collected for the y-axis was used, which denoted the vertical-to-horizontal position change when horse laid down and its metacarpal as well as tracker turned horizontal with respect to the ground.

## Disruption to experiment

During the repetition 2 habituation phase, a severe storm damaged shelters and fences, leading to evacuation of horses from the experimental site and pausing the experiment for one month and three weeks. Groups 1, 2, and 4 mares were individually housed due to logistical constraints, while group 3 remained group-housed. Mares from each group were turned out daily in small paddock groups to maintain social contact. All groups followed the SNSF feeding regime for one month and ten days (four daily forage meals at 7am, 12 pm, 4 pm, and 8 pm), followed by their assigned treatments for the final 10 days of repeat 2. Mares returned to the experimental site 11 days before data collection began, dividing the three-week habituation into two phases: 10 days at SNSF (individual feeding for groups 1, 2, and 4; group feeding for group 3) and 11 days on the experimental site, consistent with the other repetitions. Post-storm, shelters were rebuilt with the same size and structure but different orientation, resulting in slight variations in lying areas across repeats 1, 2, and 3. Finally, data collection took place from 1 to 15 March (Repeat 1), 4–17 June (Repeat 2) and 3–16 July (Repeat 3).

## Data analysis

The data files were processed using R-statistics (v. 4.3.3) within the R-Studio environment. Five hours of observations were lacking due to either a missing horse or unexpected circumstances outside of experimenters' control. Additionally,

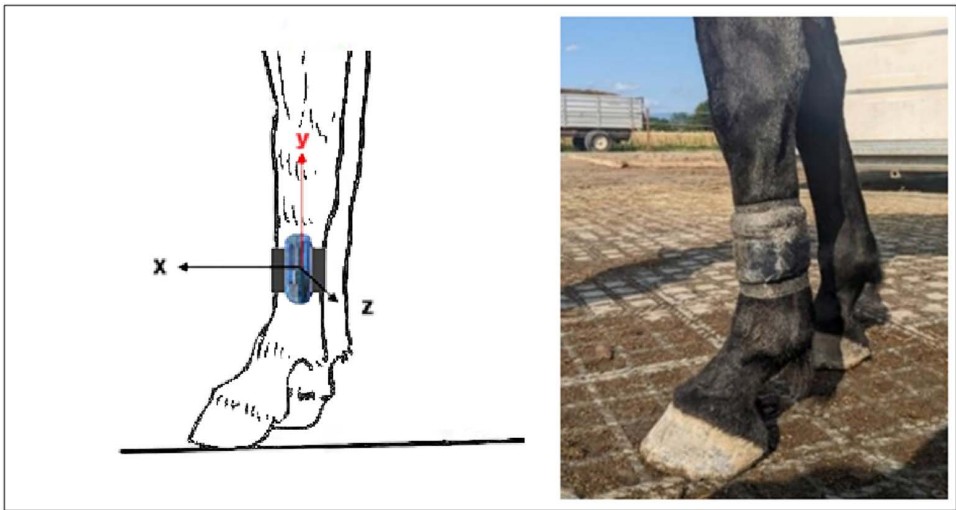

**Fig 2. Schematic representation and picture of the attachment of a MSR145 data logger to the front leg to record the lying behaviour of 18 mares during an experimental study on feeding management.**

some scan samples (activity and spatial positioning) could not be completed. Table 2 summarises the availability of videos and scans for each group and experimental repeat. Due to technical issues, the number of recording days varied between 2 and 14 days across repeats and individuals. To assess the treatment effects on response variables (social interactions, injuries, and lying behaviour), linear mixed-models were employed (see detailed descriptions in subsequent sections). Model assumptions were validated using the "check_overdispersion" function from the {fitdistrplus} package [44]. Models with multiple fixed effects (≥ 2) were evaluated based on the significance of fixed effects and AIC values. Post-hoc comparisons of treatment effects (or other significant variables) were conducted using the "pair" function from the {emmeans} package with Tukey's test [45].

**Time-budget and use of space.** The distribution of activities was assessed for each treatment group. For horses in the SF treatment, activity frequency was calculated by dividing the number of occurrences of each activity by the total number of scans. Conversely, groups in the PO and TD treatments were observed primarily during feeding slots (5 out of 15 scans, ratio of 1/3), despite having a daily feeding ratio of 1/4 (6 hours out of 24). To reflect their actual daily time budget, the average number of scans for each activity during both feeding and non-feeding periods was calculated. Their diurnal activity distribution was then estimated as follows: *% of activity over the day = 1 * (% of scans during feeding bouts) + 3 * (% of scans during non-feeding bouts)*. Regarding spatial utilisation, proportions of scans in the feeding area, lying and drinker area, and trail were calculated for each treatment. Descriptive statistics were used to analyse daily space utilisation.

**Social interactions.** We calculated the frequency of affiliative and agonistic interactions within groups for each hour of observation. To achieve this, all affiliative and agonistic interactions were normalized by dividing the total number by the number of horses in each group. This yielded the frequency of affiliative (and agonistic) behaviours per hour per horse, accounting for group size variability. Two mixed-linear models were then applied using the "lmer" function from the {lme4} package [46]: one for agonistic interactions and another for affiliative interactions. Treatments were treated as fixed effects, while repetition, observer, and group were treated as random effects. To compare social interactions during and outside of mealtimes, and considering that hay racks remained open during SF, two separate analyses were conducted. The first analysis included observations when hay racks were open (TD and PO mealtimes, and all SF observations). Due to the skewed distribution of interactions observed during SF, a log transformation (1 + log(response variable)) was applied to both agonistic and affiliative interactions. The second analysis included observations when hay racks were closed (TD and PO). Here, the response variable distribution was closer to normal, eliminating the need for transformation. Given the higher frequency of agonistic interactions observed during feeding slots in PO and TD, we examined whether this pattern persisted throughout the meal. We compared agonistic behaviour frequencies across the four quarters of the feeding slots using a linear mixed model. The same transformation (1 + log(response variable)) was applied, with treatment and quarter

**Table 2. Observation hours and number of scans used for behavioural analysis of 18 horses divided into four groups, during an experimental study on feeding management.**

| | Repeat 1 | Repeat 2 | Repeat 3 | Total number per group |
|---|---|---|---|---|
| **Group 1** | Observations = 16<br>Scans = 79 | Observations = 15<br>Scans = 69 | Observations = 14<br>Scans = 74 | Observations = 45<br>Scans = 222 |
| **Group 2** | Observations = 17<br>Scans = 74 | Observations = 14<br>Scans = 69 | Observations = 12<br>Scans = 76 | Observations = 43<br>Scans = 219 |
| **Group 3** | Observations = 17<br>Scans = 80 | Observations = 15<br>Scans = 64 | Observations = 16<br>Scans = 69 | Observations = 48<br>Scans = 213 |
| **Group 4** | Observations = 16<br>Scans = 76 | Observations = 15<br>Scans = 67 | Observations = 13<br>Scans = 74 | Observations = 44<br>Scans = 217 |
| **Total number per repeat** | Observations = 66<br>Scans = 309 | Observations = 59<br>Scans = 269 | Observations = 55<br>Scans = 293 | **Observations = 180**<br>**Scans = 871** |

(1: 0–15 min, 2: 15–30 min, 3: 30–45 min, 4: 45–60 min) as fixed effects, and repetition, observer, and group as random effects.

**Injuries.** The count data for new injuries on days +2 and +4 were analysed using a generalized linear mixed model (function "glmer" from {MASS}) with a Poisson family. Treatment was included as a fixed effect, while repeat, day, group, and horses were considered random effects (with Day nested in Repeat and Horse nested in Group). Due to high variance observed in the random component Repeat:Day, possibly due to the unforeseen event, a new model was constructed with Repeat as a fixed effect. The selection and significance of fixed effects in the best model were used to discuss the variance in random variables observed in the initial model, thereby assessing the impact of the unforeseen event. Additionally, descriptive statistics and data visualization were used to analyse the location of injuries.

**Lying behaviour.** An algorithm (written in R-statistics) was developed to detect all bouts of lying behaviour from the accelerometer data, in a similar process to that described by [23]. The algorithm also provided a summary per horse per day, including the following information: shorter bout, longer bout, mean duration of bout, number of bouts and total time spent lying down over the day. Additionally, a summary per horse was generated, including the following details: mean duration over the whole data collection period, number of nights analysed, number of bouts, min and max duration of bouts. The second summary was used to verify the likelihood of the data. A comparison of the data obtained with this algorithm and the data obtained when using the {triact} package [47] was performed to assess the reliability of the data, yielding identical results. Some horses spent short amounts of time lying down each day, resulting in a left-skewed distribution. Consequently, the same transformation was applied as for the social interaction, and the data were analysed using the "lmer" function in the same way. We included Group and Horse as nested random variables, as well as Repeat and Day, in the initial model. The treatment was included as a fixed effect. However, due to the high variance of our random components and the likelihood that this was caused by the unforeseen event, we also ran a second model using Treatment, Repeat and Group as fixed effects and Day and Horse as random effects. The chosen fixed effects in the best model were used to analyse and explain the high variability observed in the random variables of the initial model.

## Results

### Time-budget and use of space

In the slow-feeding (SF) treatment, mares spent an average of 68.2% of their day foraging, predominantly at the hay rack (66.6%). In contrast, during the portioned (PO) and traditional (TD) treatments, mares spent less than 35% of their time foraging, with more time allocated to searching. Additionally, mares in the SF treatment spent less time standing vigilant (9.9%) and resting standing (14.0%) compared to the PO and TD treatments (>19% and 40% on average). The detailed distribution of the time-budget in the three treatments is shown in Table 3.

Horses primarily used the feeding area, accounting for 86.1% of all scans (feeding and non-feeding slots included). During periods when hay racks were closed (PO and TD treatments), horses were more frequently observed on the trail or in the shelter and trough area. Nonetheless, the feeding area remained their predominant location, comprising 76.6% of non-feeding scans overall (Fig 3).

**Table 3. Distribution of the activities over the day (7 am to 7 pm) of 18 horses during the three feeding treatments of an experimental study on feeding management. The treatments were SF: slow-feeding, PO: portioned and TD: traditional. The results are presented in percentage (%).**

|  | Feeding | Searching for food | Standing (vigilant) | Walking | Resting (standing) | Resting (lying) | Social interactions | Other |
|---|---|---|---|---|---|---|---|---|
| **SF** | 66.6 | 1.6 | 9.9 | 3.8 | 14.0 | 1.3 | 1.5 | 1.3 |
| **PO** | 27.1 | 7.6 | 19.5 | 3.5 | 40.7 | 0.6 | 0.0 | 1.0 |
| **TD** | 28.3 | 7.4 | 18.4 | 3.7 | 39.4 | 0.4 | 0.6 | 1.8 |

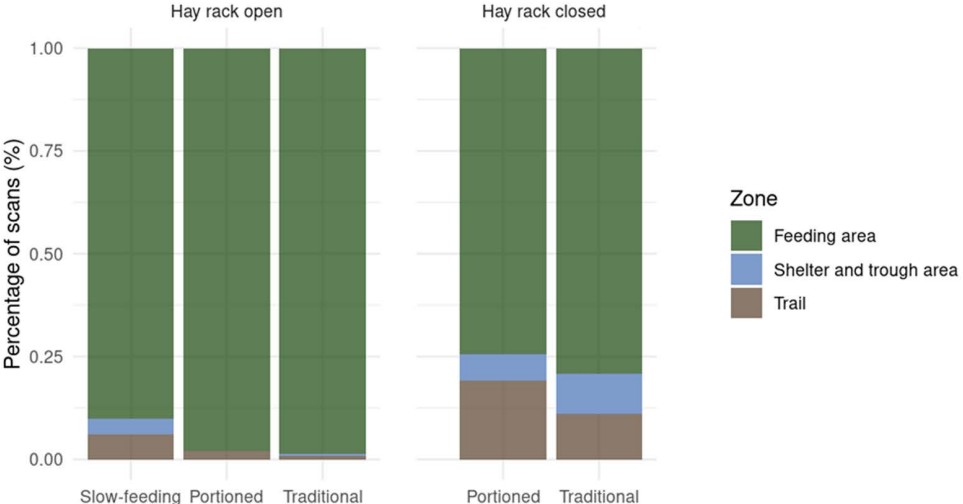

**Fig 3. Utilisation of the space across the three treatments by the 18 mares involved in a study on feeding management, depending on the hay rack status (closed or open).**

## Social interactions

**Hay racks open.** Across all treatments and when the hay racks were open, the mean number of affiliative interactions per horse and per hour ranged from 0.2 to 7.0, with 2.58 ± 1.54 affiliative interactions recorded on average. In contrast, the mean number of agonistic interactions per horse and per hour demonstrated considerable variability, ranging from 0.75 to 41.25, with an average of 10.33 ± 7.16 agonistic interactions recorded (Fig 4).

Regarding affiliative interactions, the treatment did not show a significant effect. However, for agonistic interactions, there was a significant difference observed, with a higher frequency in the PO treatment compared to SF (p = 0.0064,

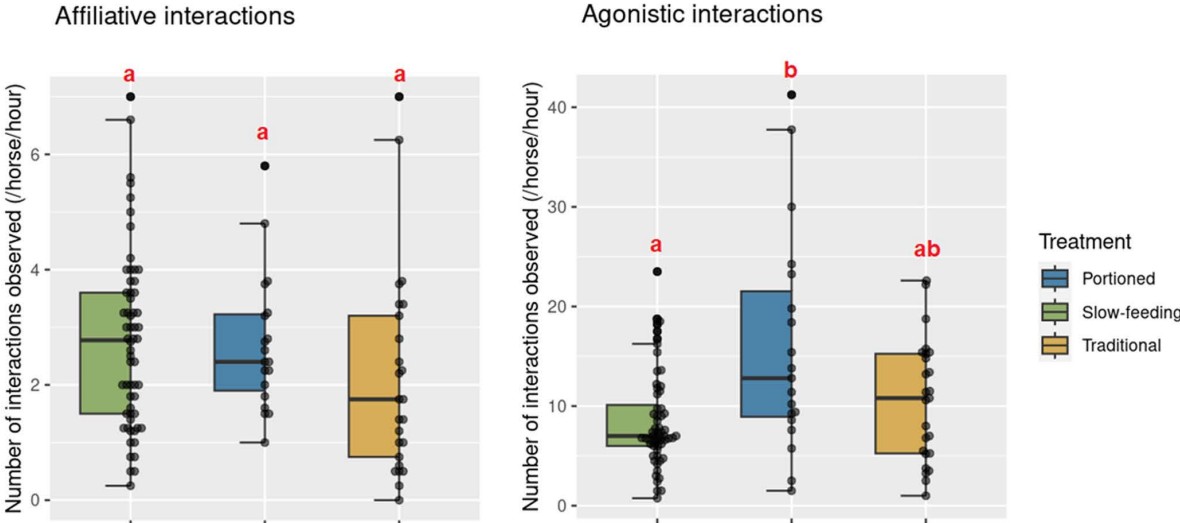

**Fig 4. Number of agonistic and affiliative interactions observed in four groups of horses in an experimental study assessing the effect of feeding management.** Agonistic and affiliative interactions are described in Table 1. The figure presents the agonistic and affiliative interactions observed during feeding times (hay racks open). Different letters indicate significant differences (post-hoc Tukey test).

post-hoc Tukey test). Conversely, no significant differences were found between TD and SF (p = 0.3347, post-hoc Tukey test) or between TD and PO (p = 0.2630, post-hoc Tukey test). The coefficients of the fixed effects and their confidence intervals, as well as the variance of the random effects for the two models are reported in Table 4.

In examining the frequency of agonistic interactions during meals (TD and PO only), the best model included the quarter-hour intervals (1st: 0–15 min, 2nd: 16–30 min, 3rd: 31–45 min, 4th: 46–60 min). Post-hoc comparisons revealed a significant increase in agonistic behaviour during the first 15 minutes compared to the subsequent quarters. Furthermore, the difference between the first and fourth quarters showed a linear increase over time, indicating a larger difference compared to the first and second quarters. The coefficients of the fixed effects and their confidence intervals, as well as the variance of the random effects for the two models are reported in Table 5.

**Hay racks closed.** In both the traditional and portioned treatments, when hay racks were closed, horses exhibited a mean of 6.02 ± 4.41 affiliative interactions per hour per horse. Conversely, they displayed a mean of 7.75 ± 4.32 agonistic interactions per hour per horse (Fig 5).

The treatment did not significantly affect either affiliative or agonistic interactions. The coefficients of the fixed effects and their confidence intervals, as well as the variance of the random effects for the two models are reported in Table 6.

### Injuries

Overall, the number of new injuries over a two-day period ranged from 0 to 25, with an average of 4.20 ± 4.60 new injuries recorded. The treatment did not significantly influence the number of injuries observed. However, there was notable variance in the random components, particularly Repeat:Day. Detailed coefficients of the fixed effects, confidence intervals, and variance of the random effects for both models are presented in Table 7. When including Repeat as a fixed effect, the best model retained only this factor. Specifically, in the first experimental repeat horses sustained significantly fewer new injuries compared to the second repeat (p = 0.0022, post-hoc Tukey test) and third repeat (p < 0.0001, post-hoc Tukey test). No significant differences were found between the second and third repeats (p = 0.4788, post-hoc Tukey test). Regarding injury locations, a majority of injuries in the TD and PO treatments occurred on the horses' bodies (49.7% and 53.8%, respectively), whereas in SF, fewer injuries were body-related (36.5%), being concentrated on the head and legs of horses instead (Fig 6).

**Table 4. Estimates of the fixed effects, with their standard errors (SE) and confidence intervals (CI) for the affiliative and agonistic interactions recorded during observations with the hay racks open in an experimental study on feeding management, conducted on 18 mares divided in four groups. The variances of the random effects are also presented, together with their standard deviations (SD). "x" indicates that the modality was used as reference in the model. Different letters in subscripts indicate significant difference between treatments (post-hoc Tukey test).**

|  | Affiliative interactions | Agonistic interactions |
|---|---|---|
| **Fixed effects (estimate ± SE) | [upper; lower 95%CI]** | | |
| Intercept | 2.74 ± 0.302[2.12; 3.38] | 2.14 ± 0.200[1.70; 2.57] |
| Treatment | | |
| Slow-feeding | x [a] | x [a] |
| Portioned | − 0.05 ± 0.390 | 0.44 ± 0.130 |
| Traditional | − 0.65 ± 0.353 | 0.19 ± 0.118 |
| **Random effects (variance ± SD) | [upper; lower 95%CI of SD]** | | |
| Group | 0.21 ± 0.460[0.00; 1.12] | 0.14 ± 0.379[0.17; 0.83] |
| Repeat | 0.00 ± 0.000[0.00; 0.50] | 0.00 ± 0.000[0.00; 0.21] |
| Observer | 0.00 ± 0.060[0.00; 0.85] | 0.00 ± 0.000[0.00; 0.30] |
| Residual | 2.19 ± 1.479[1.28; 1.69] | 0.24 ± 0.493[0.43; 0.56] |

**Table 5. Estimates of the fixed effects, with their standard errors (SE) and confidence intervals (CI) for the agonistic interactions (per 15 min) of 18 mares divided in four groups, during feeding times (hay rack open). The variance of the random effects is also presented, together with their standard deviations (SD). "x" indicates that the modality was used as reference in the model, whereas an empty row indicate that the variable was not selected in the best model). Letters in subscripts indicate significant difference between treatments (post-hoc Tukey test).**

|  | Agonistic interactions |
|---|---|
| **Fixed effects (estimate ± SE) | [upper; lower 95%CI]** | |
| Intercept | 1.49 ± 0.13 |
| Treatment | |
| Traditional | |
| Portioned | |
| Quarter | |
| 1$^{st}$ quarter | x $^a$ |
| 2$^{nd}$ quarter | − 0.31 ± 0.117 |
| 3$^{rd}$ quarter | − 0.36 ± 0.117 |
| 4$^{th}$ quarter | − 0.39 ± 0.117 |
| **Random effects (variance ± SD) | [upper; lower 95%CI of SD]** | |
| Group | 0.16 ± 0.394[0.17; 0.90] |
| Repeat | 0.05 ± 0.231[0.07; 0.71] |
| Observer | 0.00 ± 0.024[0.00; 0.45] |
| Residual | 0.27 ± 0.511[0.45; 0.57] |

## Lying behaviour

Across all treatments and repeats, horses spent varying amounts of time lying down each day, ranging from 0 to 190.77 minutes, with an average of 21.9 ± 29.36 minutes (Fig 7). Specifically, horses in SF averaged 23 minutes daily, 10 minutes in PO, and 32 minutes in TD.

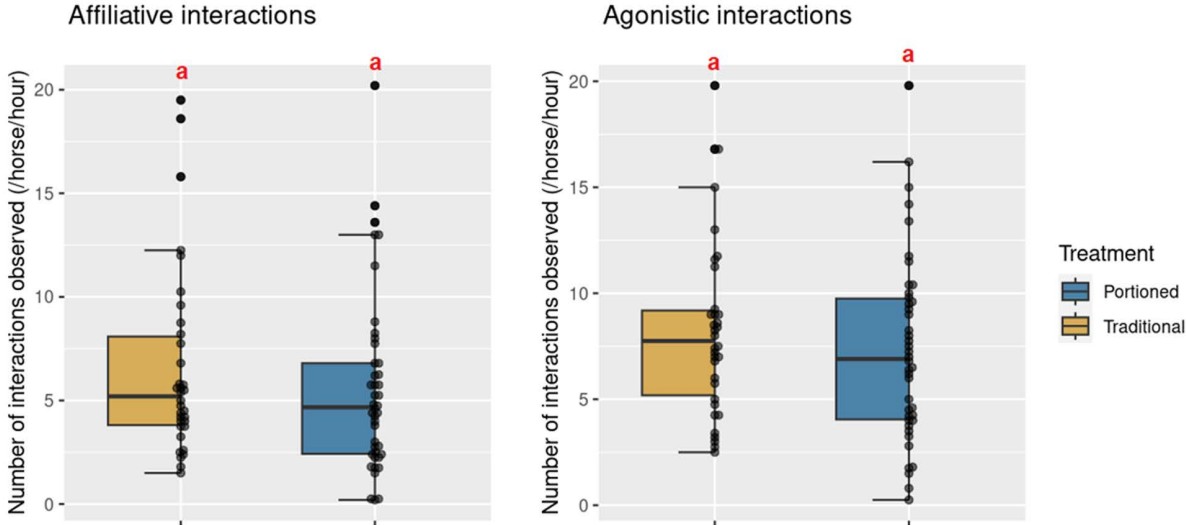

**Fig 5. Number of agonistic and affiliative interactions observed during time slots where the hay racks were closed in an experimental feeding management study conducted on 18 mares divided into four groups Different letters indicate significant differences (post-hoc Tukey test).**

**Table 6. Estimates of the fixed effects, with their standard errors (SE) and confidence intervals (CI) for the affiliative and agonistic interactions during the observations of 18 mares divided into four groups, when the hay racks were closed. The variance of the random effects is also presented, together with their standard deviations (SD). Affiliative and Agonistic interactions are described in Table 1.**

|  | Affiliative interactions | Agonistic interactions |
|---|---|---|
| **Fixed effects (estimate±SE) \| [upper; lower 95%CI]** | | |
| Intercept | 1.75±0.208[1.31; 2.20] | 8.16±1.086 [5.98; 10.36] |
| Treatment | | |
| Traditional | x [a] | x [a] |
| Portioned | 0.10±0.12[−0.14; 0.35] [a] | −0.68±0.952 [−2.62; 1.13] [a] |
| **Random effects (variance±SD) \| [upper; lower 95%CI of SD]** | | |
| Group | 0.12±0.340 [0.13; 0.78] | 2.55±1.598 [0.00; 3.83] |
| Repeat | 0.02±0.149 [0.00; 0.57] | 0.00±0.000 [0.00; 1.68] |
| Observer | 0.00±0.020 [0.00; 0.49] | 0.08±0.856 [0.00; 2.99] |
| Residual | 0.27±0.517 [0.44; 0.61] | 16.71±4.088 [3.48; 4.83] |

The treatment had a significant effect on the average time spent lying down, with horses spending significantly less time lying down during the PO treatment, compared to TD (p < .0001, post-hoc Tukey test) and SF (p = 0.0002, post-hoc Tukey test). A large variance was found for the random variables Group and Horse. The coefficients of the fixed effects and their confidence intervals, as well as the variance of the random effects for the model are reported in Table 8.

When including Repeat and Group as fixed effects, the best model included Treatment, Repeat and Group. However, only the treatments and the experimental repeats had significant effects. Horses spent significantly less time lying down in PO compared to SF (p = 0.001, post-hoc Tukey test) and TD (p < .0001, post-hoc Tukey test). They also spent significantly less time lying down during the first repeat compared to the second (p = 0.001, post-hoc Tukey test) and third repeats (p < 0.0001, post-hoc Tukey test), while the second and third repeats showed no difference (p = 0.292, post-hoc Tukey test).

**Table 7. Estimates of the fixed effects, with their standard errors (SE) and their confidence intervals (CI) for the cumulative number of new injuries recorded over 2 periods of two days for 18 mares divided into four groups in a feeding management study. The variances of the random effects are also presented, together with their standard deviations (SD). Different letters in subscripts indicate significant difference between treatments (post-hoc Tukey test).**

|  | Injuries |
|---|---|
| **Fixed effects (estimate±SE) \| [upper; lower 95%CI]** | |
| Intercept | 0.36±0.198[−0.09; 0.76] |
| Treatment | |
| Slow-feeding | x [a] |
| Portioned | 0.06±0.194 |
| Traditional | 0.16±0.188 |
| **Random effects (variance±SD) \| [upper; lower 95%CI of SD]** | |
| Group:Horse | 0.02±0.135[0.00; 0.39] |
| Repeat:Day | 0.11±0.335[0.25; 0.78] |

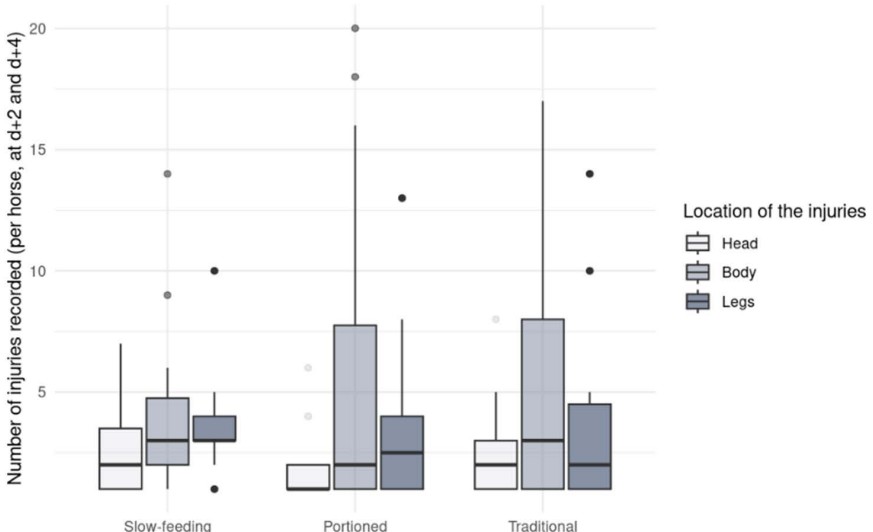

**Fig 6. Number of new injuries recorded at each location (head, body or legs) for 18 mares divided into four groups in an experimental feeding management study.** Numbers are for each horse per treatment.

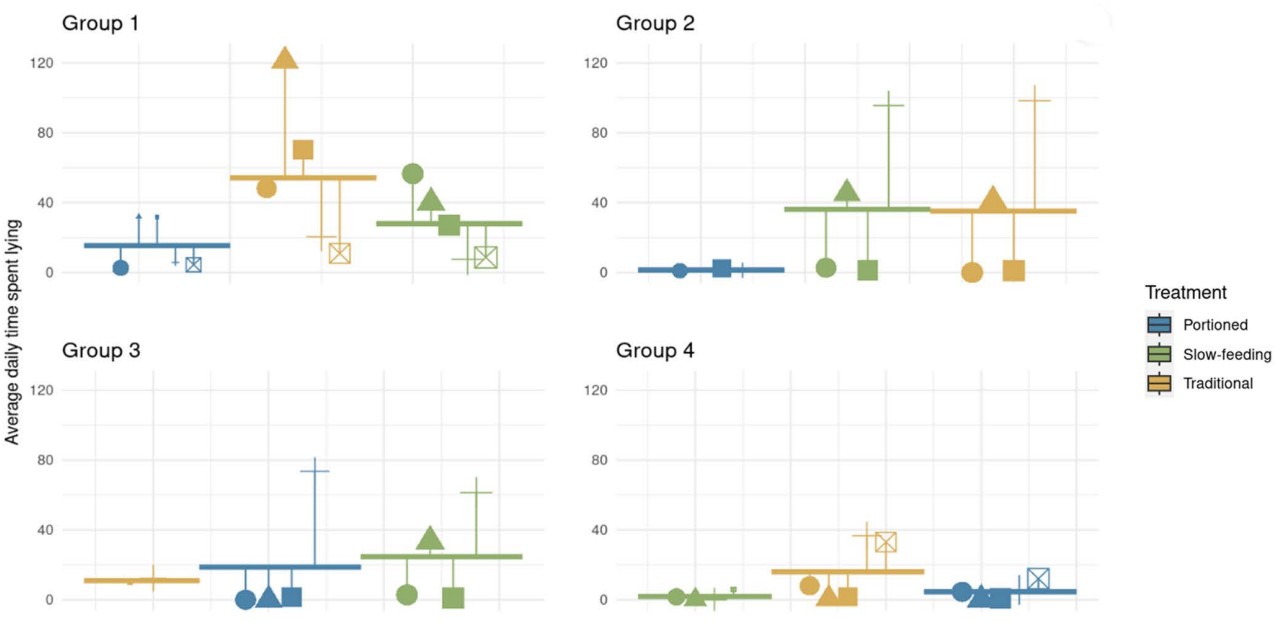

**Fig 7. Average daily lying down time (in minutes) over the three repeats of 18 horses included in a feeding management study.** The thick horizontal bar represents the group average for the treatment. Each symbol represents the average time the horse spent lying down during the treatment. Individual horse measurements with a longer vertical bar deviate more from the group average than individual horse measurements closer to the horizontal bar. The size of the symbol indicates the number of days used to estimate the horse's average (2 to 11 nights included).

**Table 8. Estimates of the fixed effects, with their standard errors (SE) and their confidence intervals (CI) for the average daily time of 18 mares divided into four groups in a feeding management study. The variances of the random effects are also presented, together with their standard deviations (SD). Different letters in subscripts indicate significant difference between treatments (post-hoc Tukey test). Please note that the response variable was transformed using (log(1+Y)) and the estimates presented are the predicted estimates without back-transformation.**

|  | Daily time spent lying |
|---|---|
| **Fixed effects (estimate±SE) \| [upper; lower 95%CI]** | |
| Intercept | 1.79±0.336[1.12; 2.46] |
| Treatment | |
| Portioned | − 0.54±0.134 |
| Traditional | 0.25±0.133 |
| **Random effects (variance±SD)** | |
| Group:Horse | 1.68±1.294[0.93; 1.83] |
| Repeat:Day | 0.37±0.608[0.24; 0.84] |
| Residual | 1.25±1.120[1.05; 1.40] |

## Discussion

Optimising feeding strategies for horses, especially those housed in groups, is crucial for enhancing their welfare and minimizing aggression-related injuries associated with hay availability. Recently, two emerging strategies aim to improve horse welfare: portioning daily intake with time-controlled hay racks and slow-feeding with nets. However, the extent of the impacts of these two strategies remain group's social dynamic understudied. We conducted a crossover study with 18 mares in four groups, comparing three feeding methods over a 5-week period: traditional (3 meals during daylight), portioned (6 meals spread over 24 hours), and slow-feeding (*ad libitum* hay via nets). Each treatment involved 3 weeks of habituation and 2 weeks of data collection, focusing on 15-hour observations per group. Social interactions were continuously recorded at 15-minute intervals, alongside injuries and lying behaviour monitored using accelerometers. Our analysis, employing generalized mixed models and AIC for model selection, revealed that portioning did not reduce horse stress, indicating a need for further research into optimal portioning strategies. In contrast, slow-feeding showed promise as a more suitable strategy, warranting additional investigation into its long-term effects on horse health and behaviour. We discuss each of our findings within the context of existing literature in the sections below.

### Time-budget and utilisation of the space

According to our hypothesis, the slow-feeding (SF) treatment facilitated a more natural time-budget. Horses in SF allocated comparable time to feeding, standing (vigilant or resting), and moving as observed in semi-feral horse populations under natural conditions [27,48–52]. Lying behaviour was infrequent (1.5% in SF), consistent with findings where recumbency phases are predominantly nocturnal [53–55]. It is important to note that our observations were limited to 7 am to 7 pm, potentially accounting for discrepancies with other studies [29,49]. Conversely, horses in TD and PO demonstrated activity time-budgets that resembled the ones observed for horses in box-stalls [56–58]. These horses spent less time feeding but more time searching for food compared to SF. The foraging behaviour observed in horses is akin to their natural search for food. However, it may also indicate compensatory feeding due to inadequate satiety from previous meals [59]. This suggests that even a two-hour feeding interval may not suffice under these treatment conditions, despite reports that grazing bouts in free-ranging horses typically last between thirty minutes to three hours [60,61].

With regard to the utilisation of space, horses spent the majority of their time in the feeding areas, even during time slots where the hay racks were closed (TD or PO). For time slots following a feeding slot, this may be explained by the horses collecting the remaining hay blades on the ground. In other time slots, this utilisation of space may be indicative

of the horses' anticipation regarding the openings of the hay racks, particularly in the absence of cues announcing the openings. In typical feeding routines, horses often anticipate meals based on cues like human presence, but the use of time-controlled hay racks removed this cue. Horses' unique spatial behaviour may also be due to hay consistently being accessible by smell, unlike conventional setups where hay is stored elsewhere. Compared to horses in the traditional feeding group (TD), those in the portioned feeding group (PO) were observed less frequently near shelters and water drinkers and more on trails, possibly indicating anticipation near hay racks. Future research should explore the effects of timed group hay racks on horse behaviour and space utilisation.

### Social interactions

**Affiliative interactions.** We surmised that slow-feeding would promote affiliative interactions within the horse groups. However, contrary to our hypothesis, we did not observe an increase in affiliative interactions with slow-feeding (SF). Affiliative interactions were limited, mainly consisting of horses following or resting near each other. Significant increases in affiliative interactions occurred when hay racks were closed in the traditional (TD) and portioned (PO) feeding groups compared to SF. This can be attributed to the fact that during feeding slots, horses were predominantly engaged in eating, which limited the opportunity of social affiliative interactions. The average frequency of agonistic interactions in our study was 10.33 per horse per hour across all treatments, which is higher than reported in similar domestic studies (range: 2.5–8.51; [21,23,28,62]. Even outside feeding times and during SF, there were approximately 7 interactions per horse per hour, which exceeds literature values. This is unexpected given the relatively stable groups with ample space per horse (340m$^2$ to 425m$^2$) and multiple feeding spots [22] noted a decrease in aggression with larger enclosures (≥331m$^2$ per horse), yet their experiment primarily involved threats and aggressive behaviours, that comprised 59.9% of the agonistic interactions in our experiment, with passive displacements making up the rest. In addition, despite the large enclosure, most mares remained within the 600m$^2$ feeding area, limiting available space (120m$^2$ to 150m$^2$ per horse).

**Agonistic interactions.** Here, we hypothesised that horses would be engaging in fewer agonistic interactions in slow-feeding (SF) compared to portioned (PO) and traditional (TD) feeding settings, and fewer during PO meals than TD. However, there were no significant differences in agonistic interactions during PO feeding slots compared to TD, contrary to expectations of reduced breaks benefiting horse welfare. Initial 15-minute periods of all feeding slots showed significantly higher aggression levels, leading to a limited access to feed for subordinate horses. Free-ranging horses typically graze 30 minutes to three hours [61], suggesting brief feeding bouts in our study may increase frustration and aggression, especially for subordinate horses, and therefore negating portioning benefits. PO's six daily feeds may also heighten emotional stress due to the increased number of food deliveries [13], reflected by higher median and variability in agonistic interactions than TD (maximum 40 vs. 22). Post-rack closure, PO and TD showed no difference in aggression, implying meal-related aggressiveness does not extend. Aggressiveness has been proposed as a potential indicator of animal welfare (review of Lesimple [63]). Consequently, the comparable level of aggressiveness observed between treatments once the hay rack is closed may indicate that the frustration or stress associated with the meals does not negatively impact the welfare of the horses outside of these slots. Future studies should explore feeding slot duration's impact on horse welfare, varying settings for ethological and digestive insights.

No significant TD vs. SF differences in feeding slot agonistic interactions were found, despite SF's *ad libitum* hay versus TD's restricted access (6 hours), contradicting our limited-forage aggression hypothesis and previous research [23]. Aggressiveness has been shown to rises with limited resources [21], while *ad libitum* hay has been found to reduce it [23,28]. The similar level of agonistic interaction between SF and TD may therefore have two potential explanations. First, a 2-hour feeding slot might allow sufficient access to hay, such that this latter is not considered a limited resource and allow for a considerable time of synchronised feeding behaviour among the group. This could be particularly pertinent in our case, given that the feeding places ratio was relatively high (1.4 to 2 feeding places per horse), granted a sufficient

access to all individuals. On the other hand, nets, and especially small hole hay nets may increase threats and aggression [29,39]. This may be due to frustration or it may indicate that hay is seen as a limited resource, thus increasing competitive interactions. However, frustration behaviours (e.g., pawing, net manipulation [35,36,64]; were very rare, but uneven net setup led dominant horses to push others for better spots. Loose hay's uniform access may explain TD's decreased aggression and lack of SF vs. TD differences.

Finally, it should be noted that in our study the impact of the feeding management was only assessed over a relatively short period of time (five weeks). This could have resulted in different outcomes than those observed in longer experiments. During TD and PO, coprophagy and wood-chewing occurred, potentially reflecting a desire to feed outside meal times due to physiological cues. These behaviours, often seen as abnormal, suggest compromised welfare [63,65]. Wood-chewing, which can precede crib-biting [66], was notably absent in SF. Such behaviours may indicate welfare issues in restrictive feeding, particularly TD, despite not immediately increasing aggression. Additionally, long fasting periods between feeding slots in restrictive management increase ulcer and other digestive problems risks [26,65]. It is therefore reasonable to assume that if the treatment had been continued for a longer period, digestive issues could have developed in TD, potentially resulting in an increased aggressiveness due to discomfort or pain. Future research is crucial to understand the long-term effects of different feeding strategies on horse welfare, so that practical management recommendations can be accurately made and the welfare of horses on farms can be ensured.

### Injuries

We expected fewer injuries in the SF treatment due to reduced social tension, potentially lowering injury risk. Although visual data analysis showed fewer injuries in SF, statistical significance was not achieved, likely due to the strong effect of repetition overshadowing treatment effects. Throughout the study, some mares exhibited atonic collapses during the day. Atonic collapses, also known as pseudo-narcolepsy episodes are associated with a lack of recumbency bouts and consists of a complete loss of muscular tonus leading to a partial or complete fall of the horse and resulting in typical skin lesions [67,68]. We observed such typical skin lesions and some mares exhibited an impaired lying behaviour, especially during PO. We can therefore assume that some mares suffered from chronical REM-sleep deprivation, which could explain the overall increase of injuries observed over the course of the experiment. However, injuries in TD and PO primarily occurred on the body (49.7% and 53.8%, respectively), likely resulting from agonistic interactions. SF, however, showed fewer body injuries (one-third of total), suggesting reduced aggressive behaviours such as kicking and biting.

### Lying behaviour

In our study we were particularly interested in the horse lying behaviour, as it has been recently suggested as a welfare indicator after several studies observed a decrease in rest/sleep in inappropriate housing [28,63,69,70]. The average daily lying down time across all treatments was low, ranging from 10 to 15 minutes, which is below reported averages in the literature (23.3 minutes to over 200 minutes in individual housing; [29,54,68,71,72]). There was a considerable inter-individual variation in daily lying times, with individual values ranging from 0 to 191 minutes across treatments. Factors such as shelter configuration (narrow with three entrances) and distance from feeding areas likely contributed to this variability. Indeed, during the day, even when hay racks were closed, horses were frequently near them (76.6% of scans). The stabilised hard ground conditions in feeding and resting areas may therefore have discouraged horses from lying down, except in a small sandy area that may provide more cushioning. As highly social animals, horses typically synchronise their activities with conspecifics [8]. However, given the limited surface of the sandy area, synchronised lying behaviour was not possible within the feeding area where horses mostly stayed.

We observed increased lying time in experimental repeats 2 and 3 compared to repeat 1, likely due to improved weather conditions transitioning from winter to summer and the introduction of soft sand areas near the shelters [68]. This

suggests that environmental changes and ground stabilization around feeding areas influenced the horses' behaviour. Additionally, the new orientation of the shelters after the storm may also be a contributing factor, as the damages indicates that the shelter may not have been adequately protecting the horses from the wind prior to this incident. Finally, the shorter time spent lying during the first repetition may indicate that horses require a longer period of habituation after being transported to a new location (longer than the two months we had in our study).

During the PO treatment, we noted impaired lying behaviour, consistent with the findings of Seabra et al. [29], who observed a reduced daily lying time (36.13 minutes) compared to ad libitum treatments (89.75 minutes and 102.47 minutes). Despite similar levels of agonistic behaviour between PO and TD, PO horses spent significantly less time lying down daily compared to TD. This discrepancy may therefore relate to night-time feeding practices, as recumbency phases are most common between 10 pm and 5 am in domestic conditions. Greening and McBride (2022) underlined that it is challenging to ascertain whether disrupted lying behaviour can be used as stress indicator or whether the disruption in lying behaviour is (partially or wholly) responsible for the animal's stress [68]. Both our and Seabra's studies incorporated feeding slots during these hours, potentially disrupting lying behaviour due to the short intervals between meals (3 hours) and the distance between feeding and lying areas. In a previous study using time-controlled hay racks with similar feeding slots (e.g., with feeding slots during the night), an impaired lying behaviour was also observed [69]. Further research is needed to fully understand the impact of feeding slot timing on horse lying behaviour. Furthermore, it is imperative to determine whether the disrupted lying down behaviour in horses that we observed throughout our study is specific to paddock-trails housing and ground stabilization methods in light of these housing conditions gaining more implementation than untreated-ground equine housing fields and paddocks.

### Limitations

Firstly, we acknowledge the limitation of this study in its exclusive focus on female horses (mares). Gender differences in horse behaviour have been noted in various studies. Campbell et al. [73] found no sex-related differences in ingestion rates, whereas Boyd [48] observed gender influencing activity budgets. Geor et al. [74] suggested that gender might impact grazing behaviour, and Górecka-Bruzda et al. [75] highlighted mares as more socially dependent than geldings. Consequently, our findings may be more relevant to groups comprised of mares, as compared to mixed or male-only groups. Additionally, using groups of four and five individuals each was suboptimal due to unequal feeding place ratios and density, which are known to affect the level of aggression [20,22,71].

Secondly, due to time constraints, only three treatments were implemented, comparing traditional feeding to emerging strategies. Including a fourth treatment with genuine ad libitum hay (without nets) would have provided insights into the precise effects of slow-feeding versus unrestricted access. This would have further elucidated the observed levels of agonistic interactions between TD and SF. Future studies should consider including all four treatments to comprehensively evaluate their impacts.

Thirdly, during experimental repeat 2, the shelter's destruction necessitated a 7-week evacuation of the horses, altering housing conditions and habituation compared to other repeats. This extended wash-out period diminished the crossover design's robustness. However, the impact on social interactions was minimal, with repeat 2 having insignificant influence on the response variables like lying down behaviour and injuries. Modifications due to shelter damage and the introduction of new sand areas further differentiated conditions across the experimental repeats throughout the study. Finally, this event postponed the time period of Repeat 2 and Repeat 3, leading to the first data collection in early Spring while the two remaining ones were conducted in Summer. This may have influenced the behaviour of the mares, possibly due to the presence of insects and the warm weather [76,77] or the cycling that is known to be impacted by the season [78].

Fourthly, the substantial variance in random effects across most response variables suggests notable differences between groups and repetitions. This variance potentially limits the external validity of our study, influenced by factors

discussed earlier. Nonetheless, we found similar impacts of the PO treatment on agonistic interactions and lying down behaviour as reported by Seabra *et al.* [29], albeit with variations in animals and housing conditions. While promising, these results underscore the importance of additional replication studies.

Lastly, slow-feeders and portioned feeding are common strategies to manage horse ingestion, although hay intake was not measured in our study [29] found slow-feeding and portioned feeding resulted in similar hay ingestion (1.9% of BW), whereas free-choice feeding exceeded 3% BW [73,79] found a similar result (2.6% of BW with nets vs. 3.2% without). Unfortunately, changes in body weight could not be accurately assessed in this study. However, both Seabra *et al.* [29] and De Boer *et al.* [73] noted differences in BW and body condition score (BCS) with and without nets, highlighting the impact of feeding strategy on horse health metrics.

## Conclusion

In conclusion, our study demonstrates that slow-feeding (*ad libitum* hay covered by a net) aligns closely with natural feeding behaviour, reduces body injuries likely caused by aggression, and lowers the frequency of agonistic interactions during feeding times compared to multiple portioned feeding (six hourly feedings). Importantly, slow-feeding did not disrupt lying behaviour, suggesting it may be a preferable strategy over multiple portioning in our experimental context. Further research is needed to confirm these findings and explore the efficacy of slow-feeding for weight management. However, our comparison of multiple portioning with traditional feeding did not reveal differences in agonistic behaviours or injuries, indicating that simply dividing daily feed into smaller meals with reduced fasting periods, without increasing forage availability, may not effectively reduce aggression. Additionally, slow-feeding showed higher agonistic interactions compared to studies with genuine *ad libitum* forage. This may suggest that the use of a net was frustrating for the horses, although frustration behaviours directed to the net very rarely observed. Future studies should assess the welfare implications of these feeding strategies, especially varying the frequency and duration of feeding sessions in portioning treatments.

## Supporting information

**S1 Table. Cross-over design used in a feeding management study, carried out on 18 mares divided into four groups.**
(PDF)

**S2 Table. Detailed ethogram used for the continuous observations.** Each behaviour (with its valence and type) is described in detail, together with its reference if necessary.
(PDF)

## Acknowledgments

We would like to express our gratitude to the interns who provided invaluable assistance with the project, in particular Emilie Bossu, for her exemplary dedication and commitment. Our gratitude also extends to all the staff of the Swiss National Stud Farm, as well as to the foundations that made the construction of the experimental site possible (Sandgrueb Stiftung, BLV -Bundesamt für Lebensmittelsicherheit und Veterinärwesen- and Stiftung ProPferd). We also deeply thank Galina Limenko for her meticulous proofreading and perceptive suggestions and comments.

## Author contributions

**Conceptualization:** Marie Roig-Pons.

**Data curation:** Marie Roig-Pons.

**Formal analysis:** Marie Roig-Pons.

**Funding acquisition:** Iris Bachmann.

**Methodology:** Marie Roig-Pons.

**Project administration:** Iris Bachmann, Sabrina Briefer Freymond.

**Supervision:** Sabrina Briefer Freymond.

**Writing – original draft:** Marie Roig-Pons.

**Writing – review & editing:** Marie Roig-Pons, Iris Bachmann, Sabrina Briefer Freymond.

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
