## [Decision Letter · Decision Letter 0]

PONE-D-25-03027Impact of feeding strategies on the welfare and behaviour of horses in groups: an experimental studyPLOS ONE

Dear Dr. Roig-Pons,

Thank you for submitting your manuscript to PLOS ONE. After careful consideration, we feel that it has merit but does not fully meet PLOS ONE’s publication criteria as it currently stands. Therefore, we invite you to submit a revised version of the manuscript that addresses the points raised during the review process.

Dear Sir,

Can you make the reviewer's suggestions to finish the process?

Best regards,

We look forward to receiving your revised manuscript.

Kind regards,

Julio Cesar de Souza, Ph.D.

Academic Editor

PLOS ONE

4. We note that Figures 1 and 2  in your submission contain copyrighted images. All PLOS content is published under the Creative Commons Attribution License (CC BY 4.0), which means that the manuscript, images, and Supporting Information files will be freely available online, and any third party is permitted to access, download, copy, distribute, and use these materials in any way, even commercially, with proper attribution. For more information, see our copyright guidelines: http://journals.plos.org/plosone/s/licenses-and-copyright.

1. You may seek permission from the original copyright holder of Figures 1 and 2 to publish the content specifically under the CC BY 4.0 license.

Reviewers' comments:

Reviewer's Responses to Questions

**Comments to the Author**

1. Is the manuscript technically sound, and do the data support the conclusions?

Reviewer #1: Yes

Reviewer #2: Yes

Reviewer #3: Yes

2. Has the statistical analysis been performed appropriately and rigorously? 

Reviewer #1: Yes

Reviewer #2: Yes

Reviewer #3: Yes

3. Have the authors made all data underlying the findings in their manuscript fully available?

Reviewer #1: Yes

Reviewer #2: Yes

Reviewer #3: Yes

4. Is the manuscript presented in an intelligible fashion and written in standard English?

Reviewer #1: Yes

Reviewer #2: Yes

Reviewer #3: Yes

5. Review Comments to the Author

Reviewer #1: This study looked at 3 feeding stragies for horses using group housed mares, assessing impacts on behaviour and injury. The manuscript is well written with excellent detail especially in the methods and declaration of confounders due to a severe weather event.

In consideration of general welfare and the impact of seasonality, please more clearly state the seasons and time periods when the study was conducted for the replicates. Especially for mares, transition to long day light under natural lighting conditions will induce cycling which could potentially alter interactions. This could be briefly touched on in the discussion or limitations.

Was body weight and body condition score monitored? Could this be reported for the groups given the differences in the feed regimes this would be of considerable interest to horse owners and welfarists. Alternatively, is there an indication of dry matter intake for the group/ individual horse and a significant change between feed regimes on this? In the limitations this wasn't discussed but it would be an important inclusion for future work if not available in this study.

Line 514 minor syntax error "due to the increased number of food delivery" Change to "Due to the increased number of food deliveries"

Reviewer #2: I really liked the article, found a interesting study and method of observations. I have some questions: How were the injuries evaluated? The horses where taken to another place or restrained? About the disruption incident did it had any impact on the data analysis? Do the authors think it might influenced in the animals feed or behaviour? Does the authors plan to do another research where they can measure the levels of cortisol in each treatment, to link it to the animals behavour?

Reviewer #3: The manuscript entitled ‘Impact of feeding strategies on the welfare and behaviour of horses in groups: an experimental study’ set out to investigate possible feeding strategies to meet the needs of horses in particular with respect to the welfare to be maintained. It is believed that the manuscript clearly presented the results obtained. The text is well presented, there are some sections that can be improved, in general it is requested to make a slight revision of the English, to shorten some sentences considered too long and to add some tables to make the text more fluent. In the introduction, the text is well written and detailed, and offers a clear explanation of the topic, there might be some details to revise especially with respect to the contextualisation of the information.

In the line 45, it would be useful to elaborate and give some more references because, despite the reduced time, social interactions are important for the horse's welfare. With regard to welfare, it might be necessary to make a brief reference to what activities horses are most involved in (e.g. training, transport and other activities that undermine their welfare). Two articles are attached in this regard:

Rizzo M. et al., Cortisol levels and leukocyte population values in transported and exercised horses after acupuncture needle stimulation Journal of Veterinary Behavior Volume 18, Pages 561 March 2017.

Piccione G. et al., Serum lipid modification related to exercise and polyunsaturated fatty acid supplementation in jumpers and thoroughbred horses Journal of Equine Veterinary ScienceVolume 34, Issue 10, Pages 1181 - 11871 October 2014.

With respect to group activities, he would emphasise and elaborate on the availability of resources. It should be made clear that not all horses are susceptible to obesity, but that there is a predisposition. In the materials and methods section, it is asked to be more specific with respect to the physical condition of the animals.

Social interactions were well evaluated, perhaps a table could be presented to summarise them, in addition to the one already presented. The results were well presented, the part referring to tables and graphs contributes well to presenting them. In the discussion section, it would be necessary to make a more specific and extensive comparison with previous studies, in order to expand the bibliography already present within the study, it would be advisable to go into more detail on competitive interactions. While the results are presented and properly discussed, it would be useful to extend the discussion on how these results may influence daily animal management practices on the farm.

6. PLOS authors have the option to publish the peer review history of their article (what does this mean? ). If published, this will include your full peer review and any attached files.

**Do you want your identity to be public for this peer review?** For information about this choice, including consent withdrawal, please see our Privacy Policy .

Reviewer #1: **Yes: ** Rachel Allavena

Reviewer #2: No

Reviewer #3: No

---

## [Author Response · Author response to Decision Letter 1]

30 Apr 2025

B. Response to the Reviewers

1. Is the manuscript technically sound, and do the data support the conclusions?

Reviewer #1: Yes

Reviewer #2: Yes

Reviewer #3: Yes

R: We thank the three reviewers for this positive evaluation of our manuscript.

2. Has the statistical analysis been performed appropriately and rigorously?

Reviewer #1: Yes

Reviewer #2: Yes

Reviewer #3: Yes

R: We thank the three reviewers for the positive evaluation of our statistical analysis

3. Have the authors made all data underlying the findings in their manuscript fully available?

Reviewer #1: Yes

Reviewer #2: Yes

Reviewer #3: Yes

R: We thank the three reviewers for this positive assessment. The datasets generated and used for the analysis have been made available on Zenodo at this DOI : 10.5281/zenodo.15090821. Access is restricted at the time of review, but the data will be made public if the manuscript is accepted for publication.

4. Is the manuscript presented in an intelligible fashion and written in standard English?

Reviewer #1: Yes

Reviewer #2: Yes

Reviewer #3: Yes

R: We would like to thank the three reviewers for their positive assessment of our work.

5. Review Comments to the Author

R: Please note that in the following answers the line numbering corresponds to that of the manuscript with the track of changes.

Reviewer #1

This study looked at 3 feeding stragies for horses using group housed mares, assessing impacts on behaviour and injury. The manuscript is well written with excellent detail especially in the methods and declaration of confounders due to a severe weather event.

In consideration of general welfare and the impact of seasonality, please more clearly state the seasons and time periods when the study was conducted for the replicates. Especially for mares, transition to long day light under natural lighting conditions will induce cycling which could potentially alter interactions. This could be briefly touched on in the discussion or limitations.

R: We deeply thank the reviewer for the very nice feedback on our manuscript. Regarding the time periods when the study was conducted, it was already stated in the first paragraph of the Material and Method “The study was conducted between February and July 2023” (L111-112). However, we took good note of the reviewer’s comment and added “Finally, data collection took place from 1 to 15 March (Repeat 1), 4 to 17 June (Repeat 2) and 3 to 16 July (Repeat 3)” (L240-241) after the description of the unforeseen event, to enhance clarity regarding the time periods.

The following sentences were included in the Limitation paragraph to address this additional limitation with transparency: “Finally, this event postponed the time period of Repeat 2 and Repeat 3, leading to the first data collection in early Spring while the two remaining ones were conducted in Summer. This may have influenced the behaviour of the mares, possibly due to the presence of insects and the warm weather [77,78] or the cycling that is known to be impacted by the season [79].” (L643-646)

Was body weight and body condition score monitored? Could this be reported for the groups given the differences in the feed regimes this would be of considerable interest to horse owners and welfarists. Alternatively, is there an indication of dry matter intake for the group/ individual horse and a significant change between feed regimes on this? In the limitations this wasn't discussed but it would be an important inclusion for future work if not available in this study.

R: We thank the reviewer for his insightful comment. Regarding body condition, it was assessed at the beginning of the experiment both by BCS and by weighting the mares. However, we decided to rely on body weight changes to optimise objectivity by weighing the mares at the end of each Repeat. Unfortunately, our scale had a 20kg margin of error and we very rarely had a weight difference > 20kg due to the very short time between two measurements. In fact, apart from the assessment after the 2nd replicate - due to the unforeseen event and the change in feeding regime independent of the study - we had 4 and 3 horses that showed a weight difference> 20kg after the 1st and 3rd replicate respectively. Therefore, the measurements were not really meaningful (due to both the margin of error and the unforeseen event) and we decided to withdraw this part of the manuscript.

To improve clarity on this issue, we have added a reference to the mares' body condition at the start of the experiment (L122-124) and a note on the problem we encountered in monitoring body weight changes (L167-171) in the Material and Methods section

Regarding the dry matted intake, this was unfortunately not feasible in our experiment, as stated in our Limitations paragraph (L655-656). In the same paragraph, we have already reported the findings of Seabra et al. and DeBoer et al. (two studies with similar treatments and protocols) regarding body weight and hay ingestion.: “Seabra et al. [29] found slow-feeding and portioned feeding resulted in similar hay ingestion (1.9% of BW), whereas free-choice feeding exceeded 3% BW. DeBoer et al. [74] found a similar result (2.6% of BW with nets vs. 3.2% without) and also noted differences in BW and body condition score (BCS) with and without nets, highlighting the impact of feeding strategy on horse health metrics.” (658-660)

Line 514 minor syntax error "due to the increased number of food delivery" Change to "Due to the increased number of food deliveries"

R: We have corrected accordingly (L527)

Reviewer #2

I really liked the article, found a interesting study and method of observations. I have some questions: How were the injuries evaluated? The horses where taken to another place or restrained? About the disruption incident did it had any impact on the data analysis? Do the authors think it might influenced in the animals feed or behaviour? Does the authors plan to do another research where they can measure the levels of cortisol in each treatment, to link it to the animals behavour?

R: We thank the reviewer for the positive feedback on our study and manuscript.

Injuries were assessed using an adaptation of the protocol described in Zollinger et al. The mares were not removed from their group, but an experimenter was lightly restrained to the mare being assessed. We have added this information to the injuries section for clarity: “This was done by one experimenter restraining a mare while a second experimenter assessed the horse. The mares were assessed in their group to avoid any stress to the horses.” (L211-213)

Regarding the effect of the disruption event, its potential impact has already been discussed in our limitation (L636-641). However, we have added a sentence to better discuss the potential impact on the mares' behaviour (feeding, aggressive behaviour and cycling). (L643-646). As for the data analysis, it was adjusted for the disruption by treating the “Repeat” variable as a fixed effect when we saw that our random components had a high variance. This is described in the Material and Methods section (L315-319).

Finally, such research is not yet planned, as this project was the final experiment of the first author's PhD. However, we hope that further research will be carried out on this topic, as we strongly believe that such a study with physiological assessment would be very valuable.

Reviewer #3

The manuscript entitled ‘Impact of feeding strategies on the welfare and behaviour of horses in groups: an experimental study’ set out to investigate possible feeding strategies to meet the needs of horses in particular with respect to the welfare to be maintained. It is believed that the manuscript clearly presented the results obtained. The text is well presented, there are some sections that can be improved, in general it is requested to make a slight revision of the English, to shorten some sentences considered too long and to add some tables to make the text more fluent.

R: We thank the reviewer for this positive comment. According to the comment, we have carefully revised the latest version of the manuscript. In addition, we would like to inform the reviewer that our previous version had been proofread by two native speakers. However, we would be willing to revise it slightly and improve the sections, if we could be told which sections is the reviewer referring to.

In the introduction, the text is well written and detailed, and offers a clear explanation of the topic, there might be some details to revise especially with respect to the contextualisation of the information.

R: We thank the reviewer for his comment. We have shortened a sentence that appeared too long (L84-85). As mentioned above, we would be happy to check these details if the reviewer could tell us which details could be improved.

In the line 45, it would be useful to elaborate and give some more references because, despite the reduced time, social interactions are important for the horse's welfare. Regarding welfare, it might be necessary to make a brief reference to what activities horses are most involved in (e.g. training, transport and other activities that undermine their welfare). Two articles are attached in this regard:

Rizzo M. et al., Cortisol levels and leukocyte population values in transported and exercised horses after acupuncture needle stimulation Journal of Veterinary Behavior Volume 18, Pages 561 March 2017.

Piccione G. et al., Serum lipid modification related to exercise and polyunsaturated fatty acid supplementation in jumpers and thoroughbred horses Journal of Equine Veterinary ScienceVolume 34, Issue 10, Pages 1181 - 11871 October 2014.

R We thank the reviewer for this comment. We think that, like the reviewer, social interactions are very important, despite their low frequency. However, we did not feel it was necessary to add any text on this subject, as we have already addressed it in line (L46-49) “Social interactions represent only a small proportion of a horse’s activity time-budget (3–4%), yet they are essential for the stability and cohesion of the equine social unit [6,7]. The presence of conspecifics provides not only opportunities for social interaction but also a sense of security, synchronization of rhythms, social learning for young horses, and protection from predators”. Given that the subject of this manuscript concerns the housing of equines, we do not consider it necessary to address the topic of the transport and use of equines. In addition, our experimental mares were not ridden nor transported. However, we would be willing to revise the manuscript accordingly if the reviewer provides us with a little more information on why he/she thinks it is necessary to address the topic.

With respect to group activities, he would emphasise and elaborate on the availability of resources. It should be made clear that not all horses are susceptible to obesity, but that there is a predisposition.

R: We agree with the reviewer that not all horses are subject to obesity, and we have indicated this in our manuscript “many horses are “easy keepers” with low energy expenditure or metabolic predispositions (…) “(L77-78). However, we do not fully understand the reviewer's request in relation to this topic. We would be willing to revise the manuscript accordingly if the reviewer clarifies his comment.

In the materials and methods section, it is asked to be more specific with respect to the physical condition of the animals.

R: We thank the reviewer for his perceptive comment. We have included a reference to the initial body condition score at the start of the study “In terms of body condition, at the start of the experiment all the mares had a BCS between 4 and 7 (i.e. they were all in perfect to overconditioned according the Henneke scale [40]).” (L122-124). Unfortunately, the BCS was only recorded at the beginning of the study as we relied on a scale to assess the effect of our feeding regimes on the physical condition of the horses. The mares were weighed at the start of the experiment and at the end of each treatment. However, unfortunately our scale had a 20kg margin of error and we very rarely had a weight difference > 20kg due to the very short time between two measurements. In fact, apart from the assessment after the 2nd replicate - due to the unforeseen event and the change in feeding regime independent of the study - we had 4 and 3 weight differences > 20kg after the 1st and 3rd replicate respectively. Therefore, the measurements were not meaningful (due to both the margin of error and the unforeseen event) and we decided to withdraw this part of the manuscript.

To improve clarity on this issue, we have a note on the problem we encountered in monitoring body weight changes (L167-171) in the Material and Methods section.

Social interactions were well evaluated, perhaps a table could be presented to summarise them, in addition to the one already presented.

R: We thank the reviewer for this comment. A table summarising the different behaviours was already included in the manuscript (Table 1). However, following the reviewer's suggestion, we have now provided a detailed ethogram in a supplementary file (S1).

The results were well presented, the part referring to tables and graphs contributes well to presenting them.

R: We thank the reviewer for the positive feedback on our Results sections.

In the discussion section, it would be necessary to make a more specific and extensive comparison with previous studies, in order to expand the bibliography already present within the study, it would be advisable to go into more detail on competitive interactions. While the results are presented and properly discussed, it would be useful to extend the discussion on how these results may influence daily animal management practices on the farm.

R: We thank the reviewer for his insightful comments.

We discussed our results in the light of the results of other studies, particularly those by Seabra and De Boer, which have similar protocols, to emphasise or discuss our findings.

We have also added a reference to another study on agonistic interaction with hay nets “On the other hand, nets, and especially small hole hay nets may increase threats and aggression [29, 38]. This may be due to frustration, or it may indicate that hay is seen as a limited resource, thus increasing competitive interaction” (L544-547).

Finally, regarding daily animal management practices on farm, as highlighted several times in the manuscript, further research is required before actual recommendation can be made, and especially research on the long-term effect of such feeding management. However, we emphasized this in a new sentence “Future research is crucial to understand the long-term effects of different feeding strategies on horse welfare, so that practical management recommendations can be accurately made and the welfare of horses on farms can be ensured.” (L562-563).

---

## [Editor Report · Decision Letter 1]

Impact of feeding strategies on the welfare and behaviour of horses in groups: an experimental study

PONE-D-25-03027R1

Dear Dr. Roig-Pons,

We’re pleased to inform you that your manuscript has been judged scientifically suitable for publication and will be formally accepted for publication once it meets all outstanding technical requirements.

Kind regards,

Julio Cesar de Souza, Ph.D.

Academic Editor

PLOS ONE
---

## [Editor Report · Acceptance letter]

PONE-D-25-03027R1

PLOS ONE

Dear Dr. Roig-Pons,

I'm pleased to inform you that your manuscript has been deemed suitable for publication in PLOS ONE. Congratulations! Your manuscript is now being handed over to our production team.

Kind regards,

on behalf of

Dr. Julio Cesar de Souza

Academic Editor

PLOS ONE